# Association of early childhood abdominal circumference and weight gain with blood pressure at 36 months of age: secondary analysis of data from a prospective cohort study

Caryl A Nowson,[1] Sarah R Crozier,[2] Siân M Robinson,[2] Keith M Godfrey,[2,3] Wendy T Lawrence,[2] Catherine M Law,[4] Cyrus Cooper,[2] Hazel M Inskip[2]

For numbered affiliations see end of article.

**Correspondence to**
Professor Caryl A Nowson;
caryl.nowson@deakin.edu.au

## ABSTRACT

**Objectives:** To assess whether changes in measures of fat distribution and body size during early life are associated with blood pressure at 36 months of age.

**Design:** Analysis of data collected from a prospective cohort study.

**Setting:** Community-based investigation in Southampton, UK.

**Participants:** 761 children with valid blood pressure measurements, born to women participating in the Southampton Women's Survey.

**Primary and secondary outcome measures:** Anthropometric measurements were collected at 0, 6, 12, 24 and 36 months and conditional changes between the time points calculated. Blood pressure was measured at 36 months. Factors possibly influencing the blood pressure were assessed using linear regression. All independent variables of interest and confounding variables were included in stepwise multiple regression to identify the model that best predicted blood pressure at 36 months.

**Results:** Greater conditional gains in abdominal circumference (AC) between 0–6 and 24–36 months were associated with higher systolic and diastolic blood pressures at 36 months (p<0.001). Subscapular skinfold and height gains were weakly associated with higher blood pressures, while greater weight gains between 0–6, 12–24 and 24–36 months were more strongly associated, but the dominant influences were AC gains, particularly from 0–6 to 24–36 months. Thus one SD score increases in AC between 0–6 and 24–36 months were associated with 1.59 mm Hg (95% CI 0.97 to 2.21) and 1.84 mm Hg (1.24 to 2.46) higher systolic blood pressures, respectively, and 1.04 mm Hg (0.57 to 1.51) and 1.02 mm Hg (0.56, 1.48) higher diastolic pressures, respectively.

**Conclusions:** Conditional gains in abdominal circumference, particularly within 6 months of birth and in the year preceding measurement, were more positively associated with blood pressure at 36 months than gains in other anthropometric measures. Above-average AC gains in early childhood may contribute to adult hypertension and increased cardiovascular disease risk.

### Strengths and limitations of this study

- This is one of few studies that have investigated detailed anthropometric changes in relation to blood pressure in early age and examined conditional changes between different age points.
- Key confounding risk factors were adjusted for in the models, including maternal education attainment and smoking during pregnancy.
- A large number of children from a cross-section of socioeconomic backgrounds were included in the study.
- We were not able to include all the children born in the course of the cohort study as blood pressure measurements were not available for all children, but the study sample was found to be similar to the larger group at 36 months of age.
- Abdominal girth at this young age may only represent a gross measure of central fat deposition and differences between individuals may represent genetically/prenatally determined differences in physique.

## INTRODUCTION

Low birth weight and rapid postnatal weight gain have been linked to increased risk of cardiovascular disease,[1] obesity and the metabolic syndrome—including hypertension[2] and insulin resistance[3]—later in life. Accelerated weight gain, characterised by above-average velocities of skeletal and non-skeletal postnatal growth, has been associated with higher blood pressure in childhood.[4] Low birth weight predicts blood pressure in later life,[5] but it is not clear how much this association can be attributed to low birth weight independently of accelerated postnatal weight gain, as infants who are born small for gestational age tend to gain weight more rapidly during the early postnatal period.[6]

It is thought that there may be critical periods at specific time points early in life when accelerated growth predisposes to hypertension later in life.[7–10] Furthermore, rapid increase in weight for length in the first 6 months has been associated with higher systolic blood pressure in 3-year-olds.[11] Few studies have assessed indicators of body fat distribution in infants and young children. Body fat distribution has been associated with risk factor scores for cardiovascular risk in young children[12] and postnatal rapid weight gain has been linked to deposition of fat centrally in children at 5 years.[6]

Therefore, insight into whether postnatal alterations in body composition influence blood pressure in early childhood is relevant to the development of preventative strategies to reduce the risk of cardiovascular disease in later life. Our aim was to assess how gains in adiposity, fat distribution and body size between birth, 6, 12, 24 and 36 months relate to the blood pressure of children at 36 months.

## METHODS
### Study sample: the Southampton Women's Survey (SWS)
The SWS is a large prospective cohort study which started in 1998.[13] A total of 12 583 non-pregnant women aged 20–34 years were recruited to the study. Detailed information on diet and sociodemographic factors was collected and children born to SWS women were assessed at birth and then followed up at home by trained research nurses. The research conformed to the principles embodied in the Declaration of Helsinki.

There were 1981 singleton live births to women in the SWS by the end of 2003. After exclusion of infants with major congenital abnormalities (n=2) and neonatal deaths (n=6), 1973 SWS infants remained for postnatal follow-up.

### Maternal and child data
When each child was 24-months-old, the occupations of its mother and her partner were recorded and the highest ranking of these used to define the child's social class. The social class scale was: Professional (I), Management and technical (II), Skilled non-manual (IIIN), Skilled manual (IIIM), Partly skilled (IV) and Unskilled (V). For 10 children whose parental occupations were missing at this time, employment status recorded during early pregnancy was used. Educational attainment of the mother recorded before pregnancy was defined in six groups, from 'none' to 'degree or above'.

### Body composition and blood pressure assessment
Anthropometric measurements were taken by trained researchers at birth, 6, 12, 24 and 36 months. Apart from those at birth, all measurements were taken in the children's homes. Infant crown-heel length was measured with a neonatometer (CMS Ltd, London, UK). Child height was measured with a portable stadiometer (Leicester height measurer; CMS Ltd). Skinfold

thicknesses were measured using Holtain skinfold callipers (Holtain Ltd) at specified sites and abdominal circumference was taken at the end of expiration using a blank tape measured against a fixed scale. Strict monitoring of the nurses' measurement techniques was performed by the senior research nurse and regular inter-observer variation studies were conducted.

Blood pressure was measured using a Critikon DINAMAP 1846 SX automated blood pressure device[14] with the child seated. Three measurements were recorded and the average of the last two used in the analysis. Owing to limited equipment availability, blood pressure measurements were only available for approximately 47% of the SWS children.

### Statistical analysis
Regression coefficients (β), with associated 95% CIs, were used to assess the strength of association between body size indicators (body weight, length/height, abdominal circumference and subscapular skinfold thickness). z-Scores were calculated for body weight and length/height using the 1990 British growth references for time points 6, 12, 24 and 36 months.[15] z-Scores for abdominal circumference and subscapular skinfolds were calculated internally using the SWS sample and were adjusted for gender, current age and gestational age. Conditional growth was derived from the residuals resulting from regression of the z-score for the measurement at a specific time point on the z-scores for measurements at all preceding ages. For example, the dependent variable 'conditional gain in body weight from 12 to 24 months' was derived as the residual of the regression of body weight z-score at 24 months on the z-scores for body weight at 12 months, 6 months and birth.

Factors reported to be associated with blood pressure were assessed using linear regression, including age, social class, maternal education attainment, smoking in late pregnancy (an indicator of smoking throughout pregnancy) and crying of the child during blood pressure measurement. Factors that were univariately associated with blood pressure at age 36 months were retained for inclusion in regression models, namely crying, smoking and education.

Multiple regression analysis was performed by entering all independent variables of interest into the model, in addition to the confounding variables. A stepwise multiple regression analysis was used to identify the growth variables that were most strongly associated with blood pressure at 36 months. Statistical analyses were performed using SPSS PASW Statistics Release V.18 (IBM SPSS, IBM Corp, New York), and Stata V.12.0 (StataCorp, Texas, USA).

### RESULTS
At 36 months of age, 1640 children (83% of the 1973 available for follow-up) were followed-up. Birth weights and blood pressure measurements at age 36 months were

available for 773 infants. Seven children with missing height and weight data at age 36 months and five with systolic pressures more than three SDs from the mean were excluded, leaving 761 in the analysis. Children in this study were similar to the larger population sample of SWS children seen at 36 months (table 1). Owing to a relative unavailability of blood pressure machines during later fieldwork, children included in the analyses were more likely to have been visited earlier in the study, and their mothers were slightly younger. Additionally, compared with those not included, those children in the study were marginally older (by approximately 1 week), and lighter and shorter at birth, and their mothers were of lower social class, had lower educational qualifications and were more likely to have been smoking in late pregnancy. The full ranges of social classes and educational levels were represented in the analysis sample, although the 'Professional/Management and technical' social class accounted for around 40% of the population, and just over half the mothers had completed higher school/post-school qualifications.

The greatest relative and absolute increases in body weight, height/length, abdominal circumference and subscapular skinfold thickness occurred between birth and 6 months (table 2), with the other age intervals (6–12, 12–24 and 24–36 months) indicating smaller positive increments for body weight and height. Mean values of height and weight were comparable to the 50th centile.[15] For subscapular skinfold, average changes between later ages from 6 m were negative, as was the average change in abdominal circumference between 24 and 36 months. During this final age period, abdominal circumference increased in 39% of children but decreased in 60%.

## Confounding variables

Age, gender and social class were not associated with blood pressure, but the 45 infants who cried during measurement had higher systolic (p=0.004) and diastolic pressures (p=0.001). Smoking in late pregnancy was associated with higher systolic (p=0.067) and diastolic pressures (p=0.005). Lower educational attainment (p=0.024) was associated with higher diastolic pressure.

## Associations with blood pressure at 36 months

Initially, the four anthropometric measurements were considered separately (table 3). Each model contained the measurement at birth and the conditional changes in the measure over the four age periods (0–6, 6–12, 12–24 and 24–36 months), along with the confounding factors that contributed significantly to the regression analysis; the slope represents the change in blood pressure (mm Hg) per SD change in growth measurement.

**Table 1** Maternal and infant characteristics of the Southampton Women's Survey study group

| | With BP measurement (n=761) | | Without BP measurement (n=879) | |
|---|---|---|---|---|
| | **Mean** | **SD** | **Mean** | **SD** |
| *Mothers* | | | | |
| Maternal age at birth of the child*** | 29.7 | 3.7 | 30.6 | 3.8 |
| Pre-pregnancy weight (kg) | 67.3 | 13.9 | 68.3 | 13.9 |
| Height (cm) | 163.0 | 6.5 | 163.4 | 6.3 |
| Maternal BMI (pre-pregnancy; kg/m$^2$) | 25.3 | 4.8 | 25.5 | 4.8 |
| | Per cent | | Per cent | |
| Smoking in pregnancy* | 16.2 | | 12.2 | |
| Social class | | | | |
| Professional/Management and technical (I/II) | 39.2 | | 43.0 | |
| Skilled manual/non-manual (III) | 48.4 | | 46.5 | |
| Partly skilled/unskilled (IV/V) | 12.4 | | 10.6 | |
| Educational attainment* | | | | |
| Compulsory education to age 16 years | 44.9 | | 39.0 | |
| Postcompulsory education | 55.1 | | 61.0 | |
| *Infants* | | | | |
| Gender—male, n (%) | 397 (52.2) | | 473 (53.8) | |
| Age in years at 36 months visit*** | 3.09 | 0.10 | 3.07 | 0.09 |
| Weight (kg) at 36 months visit | 15.0 | 1.9 | 15.1 | 1.8 |
| Height (cm) at 36 months visit | 95.7 | 3.7 | 96.0 | 3.6 |
| Abdominal circumference (cm) at 36 months visit | 51.2 | 3.2 | 51.3 | 3.1 |
| Subscapular skinfold (mm) at 36 months visit | 6.63 | 1.85 | 6.49 | 1.76 |
| Systolic BP (mm Hg) at 36 months visit | 93.8 | 8.3 | | |
| Diastolic BP (mm Hg) at 36 months visit | 58.1 | 6.3 | | |
| Birth weight (kg)** | 3.42 | 0.57 | 3.50 | 0.52 |
| Birth length (cm)* | 49.8 | 2.1 | 50.0 | 2.0 |

*p<0.05, **p<0.01, ***p<0.001.
BMI, body mass index; BP, blood pressure.

**Table 2**  Body composition at 0, 6, 12, 24 and 36 months and incremental changes

| | Mean | SD | | Absolute change Mean | SD | % Change* Mean | SD |
|---|---|---|---|---|---|---|---|
| N=712 | | | | | | | |
| Birth weight (kg) | 3.4 | 0.6 | Δwt (kg): 0–36 months | 11.6 | 1.8 | 349.7 | 92.7 |
| Weight (kg) 6 months | 7.9 | 1.0 | Δwt (kg): 0–6 months | 4.4 | 0.9 | 134.4 | 43.6 |
| Weight (kg) 12 months | 10.0 | 1.2 | Δwt (kg): 6–12 months | 2.1 | 0.6 | 27.2 | 8.2 |
| Weight (kg) 24 months | 12.5 | 1.5 | Δwt (kg):12–24 months | 2.6 | 0.9 | 26.0 | 8.5 |
| Weight (kg) 36 months | 15.0 | 1.9 | Δwt (kg): 24–36 months | 2.5 | 0.9 | 20.1 | 7.0 |
| N=666 | | | | | | | |
| Birth length (cm) | 49.8 | 2.1 | Δht (cm): 0–36 months | 46 | 3.2 | 92.7 | 8.3 |
| Height (cm) 6 months | 67.2 | 2.5 | Δht (cm): 0–6 months | 17.4 | 2.1 | 35.1 | 5.0 |
| Height (cm) 12 months | 75.6 | 2.7 | Δht (cm): 6–12 months | 8.4 | 1.8 | 12.6 | 2.8 |
| Height (cm) 24 months | 86.3 | 3.1 | Δht (cm): 12–24 months | 10.8 | 1.9 | 14.3 | 2.5 |
| Height (cm) 36 months | 95.8 | 3.6 | Δht (cm): 24–36 months | 9.5 | 1.6 | 11.0 | 1.9 |
| N=682 | | | | | | | |
| Abdominal circum.† (cm) 0 month | 31.5 | 2.2 | Δcircum. (cm): 0–36 months | 19.8 | 3.1 | 63.3 | 12.3 |
| Abdominal circum. (cm) 6 months | 47.4 | 3.2 | Δcircum. (cm): 0–6 months | 15.9 | 3.3 | 50.9 | 12.3 |
| Abdominal circum.(cm) 12 months | 49.6 | 3.2 | Δcircum. (cm): 6–12 months | 2.2 | 2.6 | 4.7 | 5.7 |
| Abdominal circum. (cm) 24 months | 51.9 | 3.4 | Δcircum. (cm): 12–24 months | 2.3 | 2.9 | 4.8 | 6.0 |
| Abdominal circum. (cm) 36 months | 51.3 | 3.1 | Δcircum. (cm): 24–36 months | −0.6 | 2.5 | −1.0 | 4.7 |
| N=645 | | | | | | | |
| Subscapular skinfold (mm) 0 month | 5.0 | 1.0 | Δsubscapular‡ (mm): 0–36 months | 1.7 | 1.9 | 37.2 | 42.4 |
| Subscapular skinfold (mm) 6 months | 7.4 | 1.6 | Δsubscapular (mm): 0–6 months | 2.4 | 1.7 | 52.7 | 41.7 |
| Subscapular skinfold (mm) 12 months | 7.2 | 1.6 | Δsubscapular (mm): 6–12 months | −0.2 | 1.5 | −1.8 | 19.3 |
| Subscapular skinfold (mm) 24 months | 6.6 | 1.6 | Δsubscap. (mm): 12–24 months | −0.6 | 1.4 | −7.3 | 19.3 |
| Subscapular skinfold (mm) 36 months | 6.7 | 1.9 | Δsubscapular (mm): 24–36 months | 0.1 | 1.4 | 3.6 | 20.1 |

*Calculated from previous time interval.
†Abdominal circumference.
‡Subscapular skinfold thickness.

In all four models, the measurements at birth were not associated with blood pressure, and independent of measurements of postnatal growth. The model for abdominal circumference explained more of the variance in blood pressure (8.8% and 7.7%, respectively, for systolic and diastolic blood pressure; figure 1) than the models for the other three measures, with the model for weight coming a close second (6.7% and 5.5% of the variance explained for systolic and diastolic blood pressure). The change in abdominal circumference closest to the blood pressure measurement, 24–36 months, was related most strongly to blood pressure (systolic and diastolic), but change during the first 6 months of life was also significantly associated with systolic and diastolic blood pressure. Weight change between birth and 6 months was related to blood pressure at 36 months, but weight change between 12 and 24 months also appeared to influence systolic and diastolic blood pressure. Neither height nor subscapular skinfold thickness changes were related to blood pressure as strongly as abdominal circumference or weight changes, although for diastolic blood pressure there was a robust association with height change between 12 and 24 months. The effect sizes for the significant associations of body fat distribution and body weight were such that a one SD score increase in the measurement between the two

ages under consideration was associated with an increase of around 1–2 mm Hg in systolic blood pressure and approximately 1 mm Hg in diastolic pressure.

The results of the final regression models are presented in table 4. In the combined model, abdominal circumference (0–6 and 24–36 months) and weight change (12–24 months) remained significantly associated with systolic blood pressure. The growth variables contributing to the final models were not highly correlated, with all correlations being less than 0.15. There was considerable variability in the gains of abdominal mass between 24 and 36 months, where 39% of children experienced an increase in abdominal circumference. Abdominal circumference change in the age period leading up to the blood pressure measurement was the key influence on blood pressure, with a one SD score change in the circumference being associated with a 1.66 mm Hg increase in systolic blood pressure. The final model for diastolic blood pressure was similar, with key associations being abdominal circumference change in the earliest and latest age periods (0–6 and 24–36 months) with a one SD increase in abdominal circumference change associated with approximately 1 mm Hg higher diastolic blood pressure. After the abdominal circumference changes were included in the model, changes in weight no longer appeared to influence blood pressure, but height change

**Table 3** Multiple regression models: associations between conditional gain (z-scores) in body composition measures and blood pressure*†

| | | Systolic pressure | | | | Diastolic pressure | | | |
|---|---|---|---|---|---|---|---|---|---|
| | Adjusted $R^2$ | β | 95% CI | | p Value | Adjusted $R^2$ | β | 95% CI | | p Value |
| Model 1—weight‡ (N=684) | 0.067 | | | | | 0.055 | | | | |
| Birth weight z-score | | 0.22 | −0.45 | 0.89 | 0.5 | | 0.12 | −0.38 | 0.62 | 0.6 |
| Weight: 0–6 months§ | | 1.40 | 0.76 | 2.02 | <0.001 | | 0.81 | 0.34 | 1.27 | 0.001 |
| Weight: 6–12 months§ | | 0.51 | −0.09 | 1.12 | 0.1 | | 0.36 | −0.09 | 0.82 | 0.1 |
| Weight: 12–24 months§ | | 1.20 | 0.59 | 1.81 | <0.001 | | 0.76 | 0.30 | 1.22 | 0.001 |
| Weight: 24–36 months§ | | 1.07 | 0.46 | 1.67 | 0.001 | | 0.44 | −0.01 | 0.89 | 0.06 |
| Model 2—length/height‡ (N=649) | 0.027 | | | | | 0.038 | | | | |
| Birth length z-score | | 0.41 | −0.36 | 1.18 | 0.3 | | 0.08 | −0.49 | 0.65 | 0.8 |
| Height: 0–6 months§ | | 0.91 | 0.27 | 1.54 | 0.005 | | 0.42 | −0.05 | 0.89 | 0.08 |
| Height: 6–12 months§ | | 0.66 | 0.02 | 1.30 | 0.04 | | 0.27 | −0.20 | 0.74 | 0.3 |
| Height: 12–24 months§ | | 0.74 | 0.09 | 1.38 | 0.03 | | 0.72 | 0.25 | 1.20 | 0.003 |
| Height: 24–36 months§ | | 0.32 | −0.33 | 0.96 | 0.3 | | 0.15 | −0.32 | 0.63 | 0.5 |
| Model 3—abdominal circumference¶ (N=664) | 0.088 | | | | | 0.077 | | | | |
| Birth abdominal circumference z-score | | 0.22 | −0.40 | 0.84 | 0.5 | | −0.29 | −0.76 | 0.18 | 0.2 |
| Abdominal circumference§: 0–6 months | | 1.59 | 0.97 | 2.21 | <0.001 | | 1.04 | 0.57 | 1.51 | <0.001 |
| Abdominal circumference§: 6–12 months | | −0.03 | −0.64 | 0.59 | 0.9 | | −0.23 | −0.70 | 0.23 | 0.3 |
| Abdominal circumference§: 12–24 months | | 0.58 | −0.03 | 1.19 | 0.06 | | 0.46 | −0.005 | 0.92 | 0.053 |
| Abdominal circumference§: 24–36 months | | 1.84 | 1.24 | 2.46 | <0.001 | | 1.02 | 0.56 | 1.48 | <0.001 |
| Model 4—subscapular skinfold thickness¶ (N=630) | 0.018 | | | | | 0.035 | | | | |
| Birth subscapular skinfold | | −0.17 | −0.85 | 0.52 | 0.6 | | −0.25 | −0.76 | 0.26 | 0.3 |
| Subscapular skinfold§: 0–6 months | | 0.35 | −0.33 | 1.02 | 0.3 | | 0.09 | −0.41 | 0.59 | 0.7 |
| Subscapular skinfold§: 6–12 months | | 0.99 | 0.32 | 1.65 | 0.004 | | 0.79 | 0.30 | 1.28 | 0.002 |
| Subscapular skinfold§: 12–24 months | | 0.52 | −0.14 | 1.17 | 0.1 | | 0.02 | −0.46 | 0.50 | 0.9 |
| Subscapular skinfold§: 24–36 months | | 0.54 | −0.12 | 1.19 | 0.1 | | 0.25 | −0.24 | 0.73 | 0.3 |

*Residuals derived from regression model with the specified z-score as the independent variable.
†All analyses adjusted for crying, maternal education and maternal smoking in late pregnancy.
‡z-Scores derived from percentile curve growth charts UK.[15]
§Adjusted for measurements at all preceding time points.
¶z-Scores derived internally from Southampton Women's Survey data.

**Figure 1** Association between change in abdominal circumference (residuals derived from regressing the z-score at each specific age on the z-score at all preceding ages, adjusted for crying and maternal educational attainment and smoking during pregnancy) as the independent variable and blood pressure at 36 months (mm Hg). z-Scores are based on data from the Southampton Women's Survey, adjusted for gender, current age and gestational age.

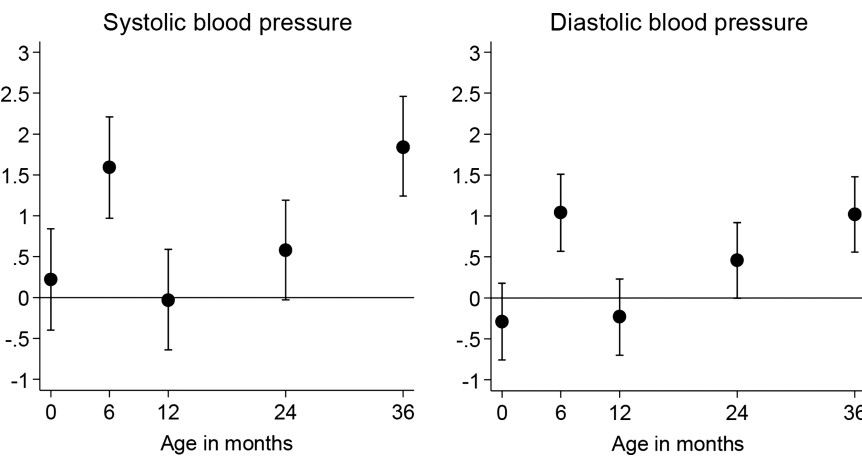

between 12 and 24 months was retained in the model. The addition of the variables 'ever breast-fed' or 'duration of breast feeding' to the final models did not affect the relationships between blood pressure and body composition measures.

## DISCUSSION
### Main findings
Our data indicate that conditional gains in abdominal circumference between birth and 6 months, as well as in the 12 months prior to blood pressure measurement, were associated with systolic and diastolic blood pressure at 36 months of age. Changes in weight were also associated with blood pressure, particularly systolic pressure, but to a lesser extent. In the multivariate regression model, a one SD gain in abdominal circumference z-score between 24 and 36 months was associated with a 1.66 mm Hg increase in systolic blood and a one SD increase between birth and 6 months with 1.56 mm Hg.

There was considerable variability in the gains of abdominal mass at the time interval most predictive of blood

pressure (24–36 months), with 39% of children experiencing an increase, suggesting the impact of environmental factors. It is possible that dietary intake and/or physical activity levels during this early period could alter abdominal circumference gain early in life, or it may be related to differences in growth trajectories,[16] which predict later obesity. Given that abdominal circumference in the majority of children decreases between the ages of 24 and 36 months, increases seen in this time period might be an early marker for problems in later life, particularly if the effect of abdominal circumference change on blood pressure amplifies with age.

It is of note that subscapular skinfold thickness change, an indicator of subcutaneous fat deposition, had much weaker associations with blood pressure than abdominal circumference, which is an indicator of abdominal or visceral fat deposition. The finding that abdominal circumference gain between 0 and 6 months was more strongly associated with blood pressure at age 36 months than gains during 6–12 and 12–24 months could reflect the proposed critical period for childhood adiposity in the first 2 months of life,[17] or could reflect gains in adiposity associated with an adverse intrauterine environment in late gestation.

**Table 4** Multivariate regression model of best fit: conditional gain* (z-scores) in body size and fat distribution associations with blood pressure at 36 months†

|  | Adjusted $R^2$ | β | 95% CI | | p Value |
|---|---|---|---|---|---|
| Systolic pressure, N=650 | | | | | |
|  | 0.094 | | | | |
| Abdominal circumference‡§: 0–6 months | | 1.52 | 0.07 | 1.71 | 0.04 |
| Abdominal circumference‡§: 24–36 months | | 1.66 | 0.94 | 2.27 | <0.0001 |
| Weight: 12–24 months‡¶ | | 0.79 | 0.16 | 1.39 | 0.01 |
| Diastolic pressure, N=625 | | | | | |
|  | 0.076 | | | | |
| Abdominal circumference‡§: 0–6 months | | 0.95 | 0.48 | 1.42 | <0.001 |
| Abdominal circumference‡§: 24–36 months | | 0.94 | 0.46 | 1.43 | <0.001 |
| Height: 12–24 months‡¶ | | 0.60 | 0.12 | 1.08 | 0.015 |

*Residuals derived from regression model with the specified z-score as the independent variable.
†All analyses adjusted for crying, maternal education and maternal smoking in late pregnancy.
‡Adjusted for all the preceding time intervals (0–6, 6–12, 12–24, 24–36 months).
§z-Scores based on sample population data from Southampton Women's Survey.
¶z-Scores derived from percentile curve growth charts UK.[15]

## Strengths and weaknesses

As summarised earlier, a strength of this study is the large sample size drawn from the general population. A post hoc power calculation (n=650, with SD of 8 mm Hg for systolic pressure) indicated there was 80% power to detect a regression coefficient of 0.88 mm Hg per one SD of an independent variable at 5% level of significance. Our study was thus sufficiently large to detect effect sizes of clinical relevance. In addition, we have assessed conditional growth, accounting for measurements at previous ages (thus ensuring independence of growth summaries between age periods), and have adjusted for relevant confounding factors.

Few studies have collected such detailed anthropometric measurements with the methodological rigour to provide indicators of fat distribution in the early postnatal period, encompassing the transition from milk only to a mixed diet at the age of 36 months.

It is clear from the data that the strongest association with blood pressure from all the anthropometric measurements is the gain in abdominal circumference; however, we acknowledge that when building a multivariate regression model of best fit, there is an element of chance as to which variables are included, even when they are not highly correlated.

A limitation of the study is the inability to measure blood pressure in all children in the cohort, although there do not appear to be noteworthy differences between those who were measured and the remainder.

## Comparison with other studies

Central adiposity and a large waist circumference have been associated with higher blood pressure in adults, but there are few reports for children. In one study, children with accelerated weight gain in the first 2 years of life were fatter and had more central fat distribution at 5 years of age than other children.[6] However, the relationship of central fat deposition specifically to blood pressure, which is only one component of the metabolic syndrome, is not clear in children. Rapid weight gain during infancy (0–6 months) but not during early childhood (3–6 years) was associated with a higher metabolic risk score at age 17 years, but the association between waist circumference and blood pressure was not statistically significant in that study.[18] Waist-for-height ratio in school children has not been found to confer additional discriminative power to BMI in some studies,[19 20] but recently weight and weight-for-height changes through infancy and childhood have been associated with blood pressure at age 10[21] and at age 9.1 and 15.5 years.[22] These studies did not consider the periods 0–6 and 6–12 months separately, nor examine change in abdominal circumference. However, similar to both reports, we found that weight changes at ages closest to those at which the blood pressure measurement was carried out were most strongly associated.

There is considerable debate as to whether there are specific age periods early in life during which weight gain

and/or fat deposition are more influential in predicting cardiovascular risk. Our findings are similar to a report that weight gain between 24 and 36 months was positively associated with systolic blood pressure.[23] The mean blood pressure values in our study of 94/58 mm Hg are similar to those reported (92/58 mm Hg),[23] as are the correlations between systolic blood pressure and current weight and height at 36 months. Similarly, we also did not find an inverse association between birth weight and blood pressure. However, in contrast we found that conditional gains in body weight between all age periods were positively associated with systolic blood pressure and a similar trend for diastolic pressure, except for the period between 6 and 12 months. The differences in results may be related to our adjustment for all previous time intervals when we determined conditional growth.

Our data support a developmental contribution to the origin of elevated blood pressure in childhood, and therefore potentially in later adulthood. Indeed, we have previously demonstrated that higher fetal liver blood flow is strongly correlated with greater fat mass at birth and at 4 years, indicating that influences on body composition operate very early in life.[24] Our data show that changes in abdominal circumference during the first 6 months and in the third year of life are associated with systolic and diastolic blood pressure. This suggests that the tracking of blood pressure, evident by later childhood, might be a consequence of under supply of conditionally essential nutrients associated with elevated fetal liver blood flow and prioritisation of fat deposition in the infant.

It is not clear whether postnatal interventions would be effective in limiting growth in abdominal circumference, and any interventions to limit abdominal circumference gains would likely also reduce the rate of growth. However, there is accumulating evidence that accelerated early growth has long-term adverse physiological effects in later life, increasing cardiovascular risk factors, including high blood pressure.[25 26] Promotion of lifestyle practices, such as exclusive breast feeding up to 6 months, later introduction of solid foods and maximising opportunities for safe physical activity, could contribute to a reduction in accelerated growth early in life and be one pathway that results in a reduction in cardiovascular disease later in life.

## CONCLUSION

We have demonstrated that although conditional gains in body weight and height during specific age periods are associated with systolic and diastolic blood pressure, as previously recognised, conditional gains in abdominal circumference (associated with central fat deposition) have the strongest association with blood pressure at 36 months of age. This is especially true for the first 6 months of life and between 24 and 36 months (ie, the year prior to blood pressure measurement). Therefore, central deposition of fat in early childhood, indicated by abdominal circumference, may contribute to an

increased risk of developing hypertension later in life. Follow-up blood pressure measurements for the study group may enable confirmation of this at a later date.

**Author affiliations**
[1]Centre of Physical Activity and Nutrition Research, School of Exercise and Nutrition Sciences, Deakin University, Geelong, Victoria, Australia
[2]MRC Lifecourse Epidemiology Unit, University of Southampton, Southampton General Hospital, Southampton, UK
[3]NIHR Southampton Biomedical Research Centre, University of Southampton & University Hospital Southampton NHS Foundation Trust, Southampton, UK
[4]MRC Centre of Epidemiology for Child Health/Centre for Paediatric Epidemiology and Biostatistics, UCL Institute of Child Health, London, UK

**Acknowledgements** The authors are very grateful to the families who participated in the research.

**Collaborators** The Southampton Women's Survey Study Group includes Patsy Coakley, Vanessa Cox, Julia Hammond, Tina Horsfall and the Research Nurses who collected and processed the data for the Survey.

**Contributors** CAN conceptualised the study, performed the analysis, contributed to the interpretation and drafted the initial manuscript. SRC assisted with the analysis and contributed to the interpretation. SMR contributed to interpretation. KMG, WTL, CML and CC reviewed and revised the manuscript. HMI conceptualised the study and designed the analysis plan, performed analysis, contributed to the interpretation and revised the manuscript. All the authors approved the final manuscript as submitted.

**Funding** This work was supported by grants from the Medical Research Council, British Heart Foundation, Arthritis Research UK, National Institute for Health Research (NIHR), Southampton Biomedical Research Centre, University of Southampton and University Hospital Southampton NHS Foundation Trust, and NIHR Musculoskeletal Biomedical Research Unit, University of Oxford. The work was also financially supported in part by the Commission of the European Community, specifically the RTD Programme 'Quality of Life and Management of Living Resources', within the 7th Framework Programme, research grant no. FP7/2007–13 (Early Nutrition Project).

**Competing interests** KMG is supported by the National Institute for Health Research through the NIHR Southampton Biomedical Research Centre.

**Ethics approval** The SWS was approved by the Southampton and South West Hampshire Local Research Ethics Committee, and participants gave written informed consent.

**Provenance and peer review** Not commissioned; externally peer reviewed.

**Data sharing statement** No additional data are available.

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
