## [Reviewer comments · BMJ Open]

Some articles will have been accepted based in part or entirely on reviews undertaken for other BMJ Group journals. These will be reproduced where possible.

ARTICLE DETAILS

TITLE (PROVISIONAL)	Association of early childhood abdominal circumference and weight gain with blood pressure at 36 months of age: secondary analysis of data from a prospective cohort study
AUTHORS	Nowson, Caryl; Crozier, S; Robinson, S; Godfrey, Keith; Lawrence, Wendy; Law, Catherine; Cooper, Cyrus; Inskip, Hazel

VERSION 1 - REVIEW

REVIEWER	Laura Howe University of Bristol, UK
REVIEW RETURNED	10-Apr-2014

GENERAL COMMENTS	1. Terminology - the authors describe the associations they observe using words such as increase and decrease, which imply stronger causal inference that can be made in an observational study. These words could be avoided and replaced with less causal alternatives. For example, 'SBP increased by xx per SDS increase in abdominal circumference' could be rephrased to 'a one SDS higher abdominal circumference was associated with xx higher SBP'.2. Were the results different if smoking was retained in the model? If so, perhaps it should be kept in the analysis, given that the 5% significance level is arbitrary and that most people would consider this an important a priori confounder.3. Was crying during measurement also associated with size/growth measurements? If not, it need not be considered a confounder.4. The word 'predictor' should be avoided, since the authors are carrying out analysis of associations rather than developing a predictive model5. In my view, it would be preferable to have the same set of confounders for SBP and DBP in table 3 rather than relying on $p < 0.05$ to determine which confounders are left in the model. This would make comparisons between SBP and DBP simpler.6. I do not think the stepwise regression adds very much to the paper. I would prefer this to be removed, and the results to concentrate on the association analyses of each anthropometric measure separately. To me it does not make very much sense to mutually adjust the different measurements and time points - this makes the resulting coefficients very difficult to interpret. The results in table 3 are nice and I don't think table 4 is necessary to tell the story.
--

REVIEWER	Bamini Gopinath University of Sydney, Australia
REVIEW RETURNED	01-May-2014

GENERAL COMMENTS	This prospective cohort study followed 761 children from birth to 36 months to determine the influence of anthropometric measures on BP. They showed that there was a positive association between abdominal circumference and systolic and diastolic BP over the 36 months. Other anthropometric measures did not show such a strong relationship with BP. 1) There seems to be a discrepancy in the number of infants available for following up on pg 5, line 31 the authors state that: '...1,973 SWS infants remained for postnatal follow-up'. But on pg 7, line 47 the authors state that: '...1,640 children (83% of the cohort) were followed-up.' Could the authors please clarify this confusion? 2) Table 1 should have an additional column specifying the p-values for differences between the group with and without BP measurements. 3) What about gestational diabetes and/or other conditions present in the mother during/ before pregnancy, these could influence the weight status and BP of the infant? Other important factors that could influence observed estimates could be parental or maternal history of hypertension and/ or maternal BP and pre/post-natal maternal weight status? Finally, what about ethnicity, there have been several studies to show that ethnicity influences both adiposity and BP, this would not have been difficult to collect this information? Surely the variables in the above should be considered in the analyses? 3) It might also be valuable to look at change on change, that is, look at the change in anthropometric measures over the 36 months with concurrent change in BP measures? 4) In the second paragraph of the Discussion, the authors state that 'It is possible that dietary intake and/or physical activity levels during this early period could alter abdominal circumference gain...' Did this study collect data on dietary intakes and physical activity levels of the children, if not then this should be listed as a limitation of the study. Also, was information collected on the age at which solids were introduced? 5) There's a possibility that some of these non-significant associations were observed in this study due to a lack of statistical power, were any power analyses done? If this study was underpowered this should be detailed in the Discussion.
---

VERSION 1 – AUTHOR RESPONSE

Reviewer Name Laura Howe

Institution and Country University of Bristol, UK

Please state any competing interests or state 'None declared': None declared

1. Terminology - the authors describe the associations they observe using words such as increase and decrease, which imply stronger causal inference that can be made in an observational study. These words could be avoided and replaced with less causal alternatives. For example, 'SBP increased by xx per SDS increase in abdominal circumference' could be rephrased to 'a one SDS higher abdominal circumference was associated with xx higher SBP'.

We have reworded this wording in the abstract, and the remainder of the text within the manuscript conforms with the above suggested wording.

2. Were the results different if smoking was retained in the model? If so, perhaps it should be kept in the analysis, given that the 5% significance level is arbitrary and that most people would consider this an important a priori confounder.

Retention of smoking in the models did not significantly affect any of the associations, though it did alter a small number of the regression coefficients by more than 10% (a commonly-used definition for confounding) so we have re-run the regression analyses for Tables 3 and 4 and included all factors that were univariately associated with blood pressure at age 36mo, namely crying, smoking and education. This did not markedly alter our results and has no impact on our conclusions.

3. Was crying during measurement also associated with size/growth measurements? If not, it need not be considered a confounder.

Although crying during measurement was not associated with any size/growth measurement, we think that it is important to retain the adjustment for "crying" as this had such a significant effect on systolic pressure: the mean systolic pressure was 3.7 mmHg in those who were crying ($P < 0.01$). It should be noted that running the analysis without adjusting for crying did not materially alter the results.

4. The word 'predictor' should be avoided, since the authors are carrying out analysis of associations rather than developing a predictive model

We are grateful for this comment which is absolutely correct, and we have replaced the words 'predictor of' with 'association with' throughout the manuscript.

5. In my view, it would be preferable to have the same set of confounders for SBP and DBP in table 3 rather than relying on $p < 0.05$ to determine which confounders are left in the model. This would make comparisons between SBP and DBP simpler.

We have re-run the analysis in line with this suggestion (see response to point 3.)

6. I do not think the stepwise regression adds very much to the paper. I would prefer this to be removed, and the results to concentrate on the association analyses of each anthropometric measure separately. To me it does not make very much sense to mutually adjust the different measurements and time points - this makes the resulting coefficients very difficult to interpret. The results in table 3 are nice and I don't think table 4 is necessary to tell the story.

We do think that the results in table 4 highlight to the reader that the association of abdominal circumference to blood pressure, particularly systolic pressure, is retained even after adjusting for growth in height and weight changes and that the highest coefficient in the multivariate model for systolic pressure is for abdominal circumference: 24-36mo. The growth variables contributing to the final models were not highly correlated, with all correlations being less than 0.15. We think that this point is not as clearly represented in the single growth model associations presented in Table 3. We have, however, removed the findings from the stepwise model from the abstract and replaced them with the results from Table 3. We have also included, in the section 'Strengths and Weaknesses', the statement "It is clear from the data that the strongest association with blood pressure from all the anthropometric measurements is the gain in abdominal circumference; however, we acknowledge that when building a multivariate regression model of best fit, there is an element of chance as to

which variables are included, even when they are not highly correlated". We do appreciate the point the reviewer is making and hope that this is an acceptable compromise. If the editor wishes us to remove Table 4 we will do so, but would prefer to retain it.

Reviewer Name Bamini Gopinath

Institution and Country University of Sydney, Australia

Please state any competing interests or state 'None declared': None declared

This prospective cohort study followed 761 children from birth to 36 months to determine the influence of anthropometric measures on BP. They showed that there was a positive association between abdominal circumference and systolic and diastolic BP over the 36 months. Other anthropometric measures did not show such a strong relationship with BP.

1) There seems to be a discrepancy in the number of infants available for following up on pg 5, line 31 the authors state that: '...1,973 SWS infants remained for postnatal follow-up'. But on pg 7, line 47 the authors state that: '...1,640 children (83% of the cohort) were followed-up.' Could the authors please clarify this confusion?

We have clarified this: "At 36mo of age, 1,640 children (83% of the 1,973 available for follow-up) were followed-up.

2) Table 1 should have an additional column specifying the p-values for differences between the group with and without BP measurements.

Thank you for this point. The p-values have been added as footnotes to table 1. Those with valid blood pressure measurements were slightly older at measurement and were born to younger mothers. They were 0.2cm longer and 0.08kg heavier at birth and there were some socio-economic differences that are noted in the table and are commented upon in the text.

3) What about gestational diabetes and/or other conditions present in the mother during/ before pregnancy, these could influence the weight status and BP of the infant? Other important factors that could influence observed estimates could be parental or maternal history of hypertension and/ or maternal BP and pre/post-natal maternal weight status? Finally, what about ethnicity, there have been several studies to show that ethnicity influences both adiposity and BP, this would not have been difficult to collect this information? Surely the variables in the above should be considered in the analyses?

These are important considerations. However, although maternal factors may influence birth weight as they precede it, it does not necessarily follow that the relationship between post-natal conditional growth and blood pressure will be influenced by maternal factors during pregnancy. Furthermore, the relationship of the maternal environment to the later health of the offspring is related to an additional set of hypotheses which is outside the scope of this paper. In this analysis we wished to evaluate the key anthropometric growth associations with blood pressure and although the impact of maternal environment on the offspring is an important field of research, we feel it is beyond the scope of this paper to evaluate the association of maternal environmental and hereditary factors in addition to anthropometric factors. Furthermore, it is worth noting that in the Southampton Women's Survey the incidences of gestational diabetes and hypertension were quite low, at 1.2% and 6.1%, respectively, and that in our study population >90% were of Caucasian origin, so these factors are unlikely to have had a major impact on the analyses.

4) It might also be valuable to look at change on change, that is, look at the change in anthropometric measures over the 36 months with concurrent change in BP measures?

Thank you for this suggestion, unfortunately we only have BP at one time point so we are unable to do this analysis.

5) In the second paragraph of the Discussion, the authors state that 'It is possible that dietary intake and/or physical activity levels during this early period could alter abdominal circumference gain...' Did

this study collect data on dietary intakes and physical activity levels of the children, if not then this should be listed as a limitation of the study. Also, was information collected on the age at which solids were introduced?

Dietary factors, as assessed by a food frequency questionnaire covering the previous 3 months at 36 months, indicated the association between a few nutrients and weight gain, e.g. total energy intake and fat intake, but neither nutrients, the use of breast feeding nor age of introduction of solids were associated with gains in abdominal circumference or blood pressure. As these data did not add to our analysis and as indicated above were similar to our previous comments around maternal factors (Point 3), we feel that exploration of associations of dietary and physical activity factors to abdominal circumference gains and blood pressure were outside the scope of this paper which focuses on identification of the associations between objective measures of growth and blood pressure.

6) There's a possibility that some of these non-significant associations were observed in this study due to a lack of statistical power, were any power analyses done? If this study was underpowered this should be detailed in the Discussion.

Thank you for this suggestion. We have included a retrospective power calculation under the section 'Strengths and Weaknesses': "A post-hoc power calculation (n=650, with SD of 8mmHg for systolic pressure) indicated there was 80% power to detect a regression coefficient of 0.88mmHg per 1SD of an independent variable at 5% level of significance. Our study was thus sufficiently large to detect effect sizes of clinical relevance." We hope that this addresses the point adequately.